# Compact Integration of TiO_2_ Nanoparticles into the Cross-Points of 3D Vertically Stacked Ag Nanowires for Plasmon-Enhanced Photocatalysis

**DOI:** 10.3390/nano9030468

**Published:** 2019-03-20

**Authors:** Vo Thi Nhat Linh, Xiaofei Xiao, Ho Sang Jung, Vincenzo Giannini, Stefan A. Maier, Dong-Ho Kim, Yong-Ill Lee, Sung-Gyu Park

**Affiliations:** 1Advanced Nano-Surface Department (ANSD), Korea Institute of Materials Science (KIMS), Changwon, Gyeongnam 51508, Korea; vtnl94@kims.re.kr (V.T.N.L.); jhs0626@kims.re.kr (H.S.J.); 2Department of Chemistry, Changwon National University, Changwon, Gyeongnam 51140, Korea; 3The Blackett Laboratory, Department of Physics, Imperial College London, London SW7 2AZ, UK; xiaofei.xiao15@imperial.ac.uk (X.X.); v.giannini@imperial.ac.uk (V.G.); s.maier@imperial.ac.uk (S.A.M.); 4Instituto de Estructura de la Materia (IEM-CSIC), Consejo Superior de Investigaciones Científicas, Serrano 121, 28006 Madrid, Spain; 5Chair in Hybrid Nanosystems, Nanoinstitute Munich, Faculty of Physics, Ludwig-Maximilians-Universität München, 80539 München, Germany

**Keywords:** 3D hybrid nanostructures, localized surface plasmon resonance, hot electrons, environmental remedy, plasmon-enhanced photocatalysis

## Abstract

The compact integration of semiconductor TiO_2_ nanoparticles (NPs) into the 3D crossed region of stacked plasmonic Ag nanowires (NWs) enhanced the photocatalytic activities through synergistic effects between the strong localized surface plasmon resonance (LSPR) excitation at the 3D cross-points of the Ag NWs and the efficient hot electron transfer at the interface between the Ag NWs and the TiO_2_ NPs. This paper explored new hybrid nanostructures based on the selective assembly of TiO_2_ NPs onto 3D cross-points of vertically stacked Ag NWs. The assembled TiO_2_ NPs directly contacted the 3D Ag NWs; therefore, charge separation occurred efficiently at the interface between the Ag NWs and the TiO_2_ NPs. The composite nanomaterials exhibited high extinction across the ultraviolet-visible range, rendering the nanomaterials high-performance photocatalysts across the full (ultraviolet-visible) and the visible spectral regions. Theoretical simulations clearly revealed that the local plasmonic field was highly enhanced at the 3D crossed regions of the vertically stacked Ag NWs. A Raman spectroscopic analysis of probe dye molecules under photodegradation conditions clearly revealed that the nanogap in the 3D crossed region was crucial for facilitating plasmon-enhanced photocatalysis and plasmon-enhanced spectroscopy.

## 1. Introduction

Semiconductor photocatalysts have been widely investigated for use in environmental science and technology applications, especially toward the degradation of organic pollutants [1,2,3,4], the photocatalytic production of hydrogen [5,6,7], and the photocatalytic reduction of carbon dioxide [8,9,10]. However, conventional semiconductor photocatalysts exhibit poor photocatalytic performances under visible light illumination (λ > 400 nm) due to their wide band gaps. For example, titanium dioxide (TiO_2_) in the crystalline anatase phase has a band gap of 3.2 eV (λ = 388 nm), and TiO_2_ semiconductor nanomaterials are active under ultraviolet (UV) light (over only 7.5% of the full solar spectrum). Around 54% of solar power falls within the visible and near-infrared regions (from 400 nm to 800 nm); therefore, significant efforts have been applied toward increasing the photoresponses of semiconductors in this spectral region and enhancing photocatalytic performances.

Recently, hybrid nanostructures containing semiconductor and plasmonic metal nanomaterials have been used to enhance photocatalytic activities over the visible range. For example, integrating TiO_2_ nanoparticles (NPs) with plasmonic nanostructures can significantly improve photocatalysis performances by enhancing the localized surface plasmon resonance (LSPR) of plasmonic nanostructures, enabling efficient hot electron transfer to TiO_2_ NPs [11,12,13,14,15,16]. At the LSPR excitation, plasmonic nanostructures can absorb and concentrate visible light at nanoscale gaps (hot spots) between metallic nanostructures, and highly energetic hot electrons can be injected into nearby TiO_2_ NPs. As a result of this plasmonic sensitization process, a wide band gap TiO_2_ material that is inactive under visible light can become active under visible light [11,12,13,14,15,16,17].

Here, we report the development of 3D hybrid nanostructures through the compact integration of TiO_2_ NPs into the 3D cross-points of vertically stacked Ag NWs. The 3D hybrid nanostructures were prepared using a simple two-step vacuum filtration process, applied first to Ag NWs and second to TiO_2_ NPs, over microfiber filters. We observed that low concentrations of TiO_2_ NPs selectively localized at the small nanogap region of the 3D crossed Ag NWs. The resulting 3D composite nanomaterials exhibited a noticeable absorption across the entire UV and visible region due to a high density of hot spots in the 3D stacked Ag NWs [18]. At optimized concentrations of the TiO_2_ NPs and Ag NWs, the photocatalytic efficiencies were measured to be 49.8% over 10 min illumination and 91.3% over 60 min illumination under standard air mass (AM) 1.5G conditions. We performed theoretical simulations of the local field enhancements in the 3D crossed regions using different wavelengths, and we systematically investigated the Raman spectral changes of organic dyes as a function of the photodegradation conditions. The results clearly revealed the crucial contribution of the hot spots (3D cross-points) to the highly enhanced photocatalytic activities and surface-enhanced Raman spectroscopy (SERS).

## 2. Materials and Methods

### 2.1. Fabrication of the 3D Hybrid Nanostructures

The hybrid nanostructures were fabricated using a two-step vacuum filtration process over a glass microfiber filter (HG00047F, HyundaiMicro, Tokyo, Japan). First, 4 mL of a 0.5 wt% Ag NWs aqueous solution (N&B Co, Ltd., Daejeon, Korea) was poured onto the microfiber filter, then 4 mL of a colloidal solution containing TiO_2_ NPs was applied onto the 3D stacked Ag NWs without breaking the vacuum. The composite substrate was dried on a hot plate at 150 °C for 3 min. The TiO_2_ NPs aqueous solution (Sigma Aldrich, St. Louis, Missouri, MO, USA) used in our experiments was a mixture of the anatase (80%) and rutile (20%) phases.

### 2.2. Photodegradation of Methylene Blue

Each substrate was immersed into a petri dish containing 10 mL of a 0.05 mM MB aqueous solution and remained on the bottom of the petri dish under illumination generated from a Xenon solar simulator lamp (AM 1.5G, 100 mW/cm^2^) as a UV-visible light source. A UV-cutoff filter was placed between the lamp and the petri dish during photocatalytic activity testing in the visible range (λ > 400 nm). The photocatalytic activities of the TiO_2_ NPs themselves were tested by mixing a TiO_2_ colloidal solution with an MB aqueous solution and measuring photodegradation using a solar simulator.

### 2.3. Characterizations

The surface morphologies of the 3D nanostructures were characterized by field emission scanning electron microscopy (FE-SEM; JSM-6700F, Joel, Tyoko, Japan) and transmission electron microscopy (TEM; JEM-2100F, Joel, Tyoko, Japan), respectively. Diffuse reflectance spectra were measured using a UV-Vis-NIR spectrophotometer (Cary 5000, Agilent Technology, Santa Clara, CA, USA). Visible light is mainly reflected by glass microfibers; therefore, the extinction spectra were converted directly from the diffuse reflectance spectra. The SERS spectra were measured using a handheld Raman spectrometer (CBEx, Snowy Range Instruments, Laramie, WY, USA) equipped with 785 nm and 638 nm lasers. The incident laser power and exposure time were 10 mW and 0.01 s for the SERS measurements, respectively.

### 2.4. Numerical Simulations

Numerical simulations were carried out using finite difference time domain analysis (FDTD) techniques. In these simulations, transverse magnetic polarized incident waves with different wavelengths (450 nm, 532 nm, 633 nm, and 785 nm, respectively) were applied to the crossed Ag NW nanostructure. The simulation was simplified by including three crossed Ag NWs (the upper and lower NWs were perpendicular to the central NW). The Ag NWs were modeled as five-sided equilateral pentagons, and the radius of the pentagon was fixed at 25 nm. The mesh size was set to 0.2 nm around the junction of the crossed Ag NWs. Perfectly matched layer (PML) formulation was applied in all directions. The complex permittivity of Ag was adopted from Reference [19].

## 3. Results and Discussion

### 3.1. Compact Integration of the TiO_2_ NPs into the 3D Cross-Points of Vertically Stacked Ag NWs

3D hybrid TiO_2_/Ag NW structures were fabricated using a simple two-step vacuum filtration method. First, 3D-stacked Ag NWs were prepared by vacuum filtering 4 mL of a dispersion containing 0.5 wt% Ag NWs over a glass microfiber filter [20,21]. Then, 4 mL of an aqueous solution containing TiO_2_ NPs were poured over the 3D stacked Ag NWs without breaking the vacuum. Figure 1a shows a scanning electron microscopy (SEM) image of the 3D stacked Ag NWs. Within the 3D multilayered Ag NWs, small nanogaps formed at the 3D cross-points at which the NWs were vertically stacked. Transmission electron microscopy (TEM) images clearly revealed that each Ag NW assumed a pentagonal shape, and small nanogap regions formed at the 3D cross-points, at which the lower and upper layers were closely spaced (dotted rectangle in Figure 1b). As small pores formed in the 3D crossed region, 22 nm (average diameter) TiO_2_ NPs were selectively filtered during the vacuum filtration process. Figure 1c,d show the 3D hybrid nanostructures consisting of TiO_2_ NPs and 3D Ag NWs upon application of a 0.002 wt% TiO_2_ NP solution to the 3D stacked Ag NWs. The small TiO_2_ NPs were mainly filtered from the solution and remained on top of the 3D cross-points regions, resulting in the compact integration of TiO_2_ NPs into the 3D hot spot region. These interfaces enhanced the interactions between Ag NWs and the TiO_2_ NPs. The 3D stacked AgNWs could be also considered as photocatalytic membranes [22,23]. In addition, the compact hybrid nanostructures embedded on glass microfiber filter would be easily separated from treated solution, which can be applied in water treatment [24,25]. The TiO_2_ NPs covered the entire 3D surface of the Ag NWs when a 0.01 wt% TiO_2_ NP aqueous solution was used (Appendix A).

### 3.2. Optical Properties and Local Field Enhancements of the 3D Hybrid Nanostructures

Figure 2 presents the extinction spectra of 3D plasmonic Ag NWs and 3D TiO_2_/Ag hybrid nanostructures prepared with different TiO_2_ concentrations (see Appendix A for the extinction spectra of the bare microfiber filters and TiO_2_ NPs). The microfiber substrate deposited using 0.01 wt% TiO_2_ NPs exhibited a very low extinction in the visible region but a very high UV extinction below 400 nm. These results indicated the presence of UV light–TiO_2_ semiconductor interactions near the band gap of crystalline TiO_2_ [26]. The green tangent line indicates 388 nm, the band gap energy (3.2 eV) of the anatase TiO_2_ (Appendix A). The 3D Ag NWs exhibited strong interactions from the UV to the visible region (black line in Figure 2). The highest extinction near 326 nm was attributed to an interband transition (3.8 eV) of Ag. The 3D stacked Ag NWs also had a high extinction across the entire visible wavelength range. The 3D hybrid nanostructures possessed higher extinction than the 3D stacked Ag NWs in visible region. The high extinction of the 3D hybrid nanostructures agreed well with the high intensity of visible light under standard AM 1.5G solar illumination (navy line in Figure 2). Therefore, the 3D hybrid nanostructures were expected to increase the photocatalytic activities through enhanced light–matter interactions.

Another prominent property of the 3D plasmonic or hybrid nanostructures is the local field enhancement due to a coupled LSPR effect at the 3D cross-points [27,28,29,30,31]. Figure 3 shows the near-field enhancement distribution contours of orthogonal Ag NWs (similar to the Ag NW stack indicated by the white dotted area in Figure 1b) for different incident wavelengths obtained from finite-difference time-domain (FDTD) simulations (Appendix A). The strong field was localized and confined at the nanogap region of the cross-points of vertically stacked Ag NWs, regardless of the wavelength of incident light. The field intensity enhancement could be further increased by increasing the curvature of the metal nanostructures due to a lightning rod effect, indicating that the pentagonal shape provided stronger enhancement than the circular shape [32]. The maximum field enhancement was obtained at the bottom corners of the central pentagonal Ag NW. The maximum field enhancement increased by an order of magnitude at shorter wavelengths (Table 1). The average field enhancement along the middle Ag NW surface could be extracted from the simulation results (Table 1). Note that we extracted the field enhancement value 0.5 nm away from the Ag NW surface to avoid any staircase effects in the simulation (green lines in Appendix A). The average field enhancement corresponded to the maximum field enhancement (Table 1). The numerical simulations suggested that a highly enhanced local field could be generated at the cross-points of the 3D stacked Ag NWs. Therefore, it is important to deposit TiO_2_ NPs selectively onto these hot spot regions to ensure efficient charge separation and enhanced photocatalysis.

### 3.3. Photocatalytic Performance

The photocatalytic activities of various nanomaterials were examined using the photodegradation of an organic dye (methylene blue (MB)). The photocatalytic degradation was carried out under AM 1.5G (or 1 sun) simulated illumination with an intensity of 100 mW/cm^2^, which are standard light illumination conditions for measuring solar cell efficiency [33]. This standard light illumination provided a quantitative photocatalyst performance comparison, unlike other light sources, such as high-pressure mercury lamps with a high power [34]. All photocatalysts were independently examined in the presence of 10 mL of a 0.05 mM MB aqueous solution under 10 min illumination. Figure 4a shows the absorbance changes of an MB solution after 10 min light illumination of each photocatalytic substrate. The main absorbance peak of MB, at 665 nm, was found to decrease in intensity after light illumination of each photocatalyst. Negligible MB degradation was observed during 10 min illumination in the absence of a substrate, indicating that MB itself did not decompose under illumination. Interestingly, the 3D stacked plasmonic Ag NWs exhibited higher photocatalytic activities compared to the TiO_2_ NPs, primarily due to LSPR-induced hot electron generation and the direct decomposition of MB [35,36]. The 3D hybrid nanostructures composed of 0.002 wt% TiO_2_ NPs and 3D stacked Ag NWs showed the highest solution MB absorbance changes in the UV-visible and visible ranges (Figure 4b). At higher concentrations (i.e., 0.01wt% TiO_2_ NP solution) of TiO_2_, the photocatalytic activities decreased due to inefficient charge separation at the Ag NWs and TiO_2_ interface and blockage of MB molecular mass transport into the hot spot regions by the large surface coverage of TiO_2_ NPs (Appendix A). The photocatalytic activities of the different nanostructures were examined by calculating the decrease in MB absorbance after illumination (Table 2). The TiO_2_ photocatalyst was active only under UV illumination due to its large band gap. No MB degradation was observed under visible illumination in the presence of the TiO_2_ photocatalyst (Appendix A). The stacked 3D Ag NWs showed better activities in both the UV-visible and visible ranges compared to the TiO_2_ NPs. The optically excited plasmonic nanostructures activated the chemical transformations directly on their surfaces [37,38,39,40]. Among the TiO_2_/Ag NW composite nanomaterials, the composite prepared with 0.002 wt% TiO_2_ NPs exhibited the best MB photocatalytic activities under both spectral ranges. In the presence of the 3D hybrid nanostructures, the MB concentration degraded to one-half of its original value after only 10 min illumination.

The 3D composite nanostructures composed of 0.002 wt% TiO_2_ and 3D Ag NWs were examined in time-dependent MB degradation studies under different light illumination conditions. The changes in the UV-visible spectra during photodegradation are shown in Figure 5. The main absorption peak of MB at 665 nm decreased markedly with the irradiation time (up to 60 min) and almost vanished after 90 min (Figure 5a). After 90 min of irradiation, the blue color of the initial MB solution became transparent due to the photodecomposition of MB molecules. We observed almost the same photocatalytic performance in the presence of the 3D composite under visible light illumination (λ > 400 nm), as shown in Figure 5b. The 3D composite performed effectively under both full spectral and visible spectral (λ > 400 nm) illumination.

The photocatalytic efficiencies of the 3D composite nanomaterials are illustrated in Figure 6. *C*_0_ is the initial concentration of MB, and *C* is its concentration after each irradiation time. About 90% of the initial MB concentration was photodegraded under UV/visible and visible illumination after 60 min. The rate of MB degradation decreased after 60 min. This decrease was attributed to the fact that the substrate was placed at the bottom of a petri dish, and the MB molecules in the aqueous solution were randomly dispersed throughout the 10 mL volume during illumination. The dilution of intact MB molecules decreased the rate of collision between the organic molecules and the active site. The diffusion-limited behavior of the dye molecules required longer illumination times to achieve total decomposition of the organic dyes. Due to the diffusion-limited behavior, we divided the photocatalytic kinetics into two stages: before 60 min and after 60 min. The photodegradation process was assumed to follow Langmuir–Hinshelwood kinetics, *ln* (*C*/*C*_0_) = −*kt*, where *C* is the MB concentration as a function of the photodegradation time (*t*), *C*_0_ is the initial concentration, and *k* is the reaction rate constant (insets of Figure 6) [41,42]. The rate constant obtained under UV-visible illumination was measured to be 3.893 × 10^−2^ min^−1^ during the initial stage and decreased to 1.438 × 10^−2^ min^−1^ after 60 min (Inset of Figure 6a). Under visible light illumination, the values of *k* were found to be 3.876 × 10^−2^ min^−1^ and 1.084 × 10^−2^ min^−1^, respectively (inset of Figure 6b).

### 3.4. Mechanism Underlying the Plasmon-Enhanced Photocatalysis

Semiconductor TiO_2_ NPs are typically integrated with 3D plasmonic Ag NWs with the goal of using the LSPR properties of 3D plasmonic Ag NWs to enhance visible light absorption and channel hot electrons to the TiO_2_ NPs for efficient photocatalysis, a plasmonic sensitization process [17]. In 3D hybrid nanostructures, a Schottky barrier forms at the junction interface between Ag and TiO_2_ (Figure 7). This barrier blocks electron transfer from the Ag NWs to the TiO_2_ NPs. Upon excitation of the LSPR, highly energetic electrons (hot electrons) have sufficient energy to overcome the Schottky barrier, and they can inject into the conduction band of the TiO_2_. The plasmonic sensitization process shuttles extra electrons to the wide-band gap (3.2 eV) TiO_2_, which is originally inactive under visible light. The TiO_2_ can then perform catalytic reduction reactions [5,17]. The holes left behind in the Ag NWs can be used to oxidize organic molecules. As most of the electron/hole pairs are produced in the field-enhanced region of the 3D crossing points of vertically stacked Ag NWs (Figure 3), efficient charge separation occurs at the interface between the 3D cross-points and nearby TiO_2_ NPs. Therefore, the compact integration of crystalline TiO_2_ NPs into the 3D cross-points can enhance the photocatalytic performance over the full spectral range.

### 3.5. Spectroscopic Investigation of Plasmon-Enhanced Photocatalysis

We collected the Raman spectra of organic dye molecules to investigate their plasmon-enhanced photocatalytic degradation by 3D stacked Ag NWs and 3D hybrid nanostructures. First, 3 µL of a 5 µM MB aqueous solution was applied to each photocatalytic substrate. After drying, SERS spectra were obtained using a portable Raman spectrometer [21,43]. The SERS measurements were collected using a low-power 10 mW illumination beam with a spot size of 20 μm incident on each substrate for a very short laser exposure time of 0.01 s to avoid photocatalytic degradation of the MB molecules. The laser was then directed onto the same spot on the substrate for 10 s to induce photocatalytic degradation of the MB molecules. SERS measurements were performed to detect any SERS intensity changes in the probe molecules. Each cycle consisted of a photodegradation process (10 s illumination) and a SERS measurement (0.01 s illumination of 10 mW laser). Figure 8 presents the SERS intensity changes in MB molecules on the 3D stacked Ag NWs (Figure 8a) and composite substrate (Figure 8b) as the cycle time was increased with laser illumination at 638 nm and 20 mW. Prior to the photodegradation process, a high-intensity characteristic MB Raman spectrum was obtained from the 3D stacked Ag NWs (black line in Figure 8a) using 10 mW 638 nm laser illumination over 0.01 s. Because the 3D stacked Ag NWs had a high density of hot spots (crossed regions between Ag NWs) for use in SERS [18,21,27,44], we obtained highly enhanced MB SERS spectra from the 3D Ag NWs. The SERS intensity of MB decreased significantly after the first photodegradation process (10 s illumination at 20 mW, red line in Figure 8a). The SERS intensity remained unchanged during additional photodegradation cycles. We performed these cyclic photodegradation and SERS measurements using a 3D composite substrate (Figure 8b). Interestingly, the initial SERS intensity of the MB obtained from the 3D composite substrate increased beyond 40%, compared to the spectrum obtained from the 3D stacked Ag NWs. This increase was attributed to the compact integration of 3D TiO_2_ NPs on the crossed regions of the Ag NWs. Because 3D volumetric hot spots formed, more MB molecules could be deposited onto the TiO_2_ NPs and Ag NWs within the 3D hot spot volume [43]. The MB SERS intensity decreased significantly after only one photodegradation process (10 s illumination at 20 mW, red line in Figure 8b). Plasmonic metal nanostructures strongly absorb the light, leading to significant local heating associated with the resonant plasmonic excitations [45,46]. This thermoplasmonic effect, therefore, partially enabled the plasmon-induced reactions, particularly in the cross-point regions of Ag NWs [47].

Figure 8c presents the overall SERS intensity changes at 1620 cm^−1^ of MB molecules under different illumination conditions. Figure 8c reveals several important characteristics of the plasmon-enhanced photocatalysis reaction. First, the 3D stacked Ag NWs displayed higher photocatalytic activities at shorter illumination wavelengths, in agreement with simulation results of local field enhancements and the optical extinction spectra of 3D plasmonic Ag NWs. Second, for a given photocatalyst and incident wavelength of light, increasing the incident light intensity increased the photocatalytic activity [48]. These results indicated that a quantitative comparison of the photocatalytic activities of the different catalysts required fixed light irradiation conditions, including the light source and intensity. Third, the 3D composite structures exhibited superior photocatalytic activities over the 3D Ag NWs, in good agreement with the photocatalytic performances. Finally, the photodegradation process mainly occurred at specific sites. We clearly observed that the SERS intensities of the probe molecules dramatically decreased only during the first cycle, then remained unchanged during the additional nine cycles. This result was attributed to the fact that prior to the photodegradation process, MB molecules deposited onto the crossed regions mainly accounted for the high SERS intensity. MB molecules on the hot spots decomposed during the first photodegradation process, and the SERS intensity decreased significantly due to the absence of target molecules in the hot spot. MB molecules deposited onto the non-crossed regions did not decompose, even after ten cycles, and MB molecules in non-hot spot regions (such as Ag NWs at the non-crossed region) contributed to a low SERS intensity. That is, hot spots were cross-points in the 3D stacked Ag NWs and hybrid TiO_2_/Ag NWs structures for the SERS and photodegradation process. The spectroscopic studies of plasmon-enhanced photocatalysis revealed that excess irradiation onto hot spot areas could trigger the photo-induced elimination of target molecules, reducing the sensitivity (or increasing the limit of detection) in highly sensitive plasmonic nanostructures during SERS measurements.

## 4. Conclusions

In this paper, we proposed the use of new composite nanomaterial into which TiO_2_ NPs were compactly integrated into the 3D cross-points of vertically stacked Ag NWs for plasmon-enhanced photocatalysis. The composite nanomaterials improved the photocatalytic activities under UV-visible and visible illumination due to the synergistic effects of visible light absorption by the 3D Ag NWs and efficient charge separation at the interface between the Ag NWs and TiO_2_ NPs. We performed theoretical simulations of the local field enhancements by the 3D stacked Ag NWs illuminated at different wavelengths, and we systematically investigated the Raman spectra during plasmon-enhanced photocatalysis. These results revealed that the organic dyes underwent photo-induced decomposition mainly at the cross-points of the 3D vertically stacked Ag NWs and 3D hybrid nanostructures. As the photocatalytic activities were highly localized at specific regions of the 3D nanostructures, it is important to enhance the mass transport of reagents to the hot spot regions to boost the photocatalytic performance.

## Figures and Tables

**Figure 1 nanomaterials-09-00468-f001:**
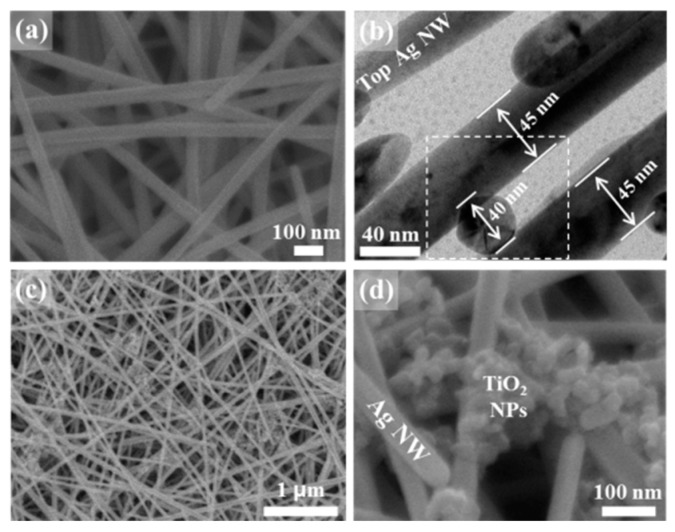
(**a**) SEM and (**b**) TEM images of 3D stacked Ag NWs. (**c**,**d**) SEM images of 3D hybrid nanostructures consisting of TiO_2_ NPs and 3D Ag NWs. The image clearly shows the compact integration of TiO_2_ NPs onto the crossed region of the 3D Ag NWs.

**Figure 2 nanomaterials-09-00468-f002:**
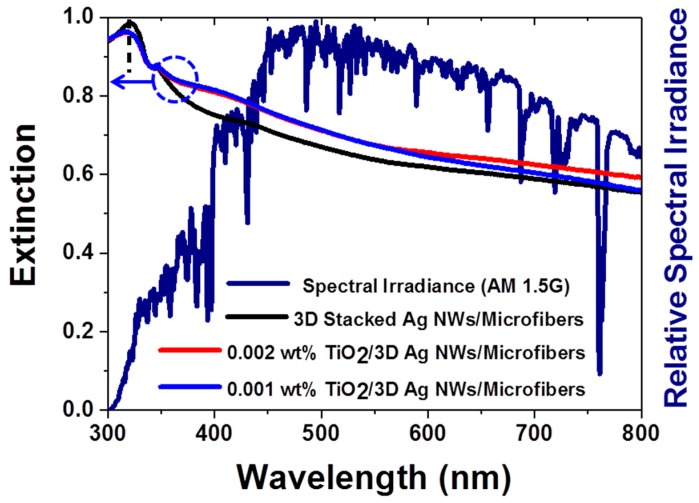
Extinction spectra of different nanomaterials and spectral irradiance of AM 1.5G. The composite nanostructures displayed high extinction in the visible range, in agreement with the solar spectrum in the visible range. The black lines indicate the interband transitions of Ag.

**Figure 3 nanomaterials-09-00468-f003:**
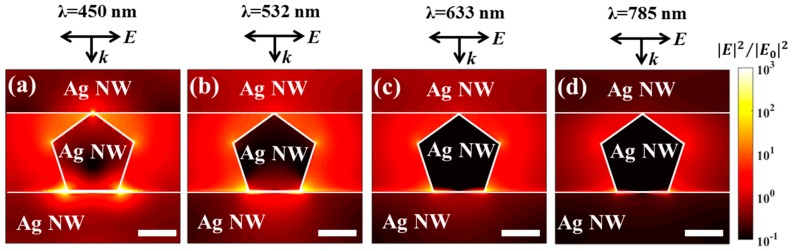
Finite difference time domain (FDTD) simulations of the electric field distribution (|E|^2^/|E_0_|^2^) at the junctions of crossed Ag NWs, using the wavelengths (**a**) 450 nm, (**b**) 532 nm, (**c**) 633 nm, and (**d**) 785 nm. The k and E vectors in the figure show the incident direction and the polarization direction of the illumination, respectively. The white lines are used to indicate the boundaries of Ag NWs. All scale bars indicate 20 nm.

**Figure 4 nanomaterials-09-00468-f004:**
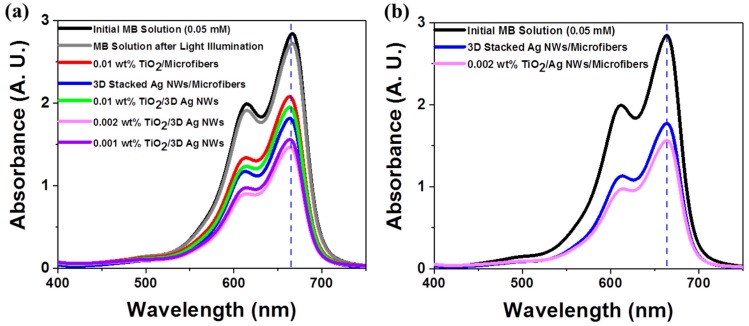
Photocatalytic performances of 0.05 mM methylene blue (MB) aqueous solutions after 10 min light illumination in the presences of different photocatalysts under (**a**) AM 1.5G simulated illumination and (**b**) visible and near-infrared illumination (λ > 400 nm).

**Figure 5 nanomaterials-09-00468-f005:**
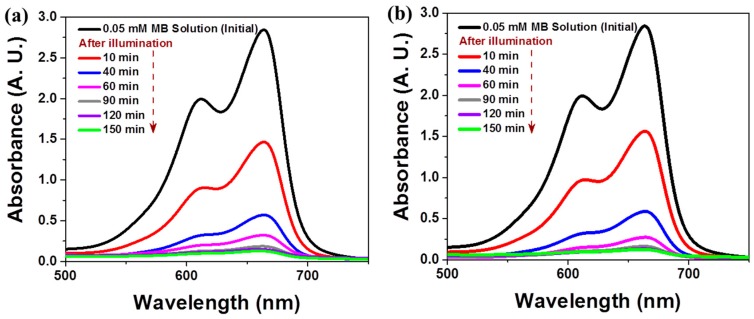
Time-dependent UV-vis spectra of an MB aqueous solution after illumination in the presence of 0.002 wt% TiO_2_/Ag NW composites under (**a**) UV-visible light and (**b**) visible light.

**Figure 6 nanomaterials-09-00468-f006:**
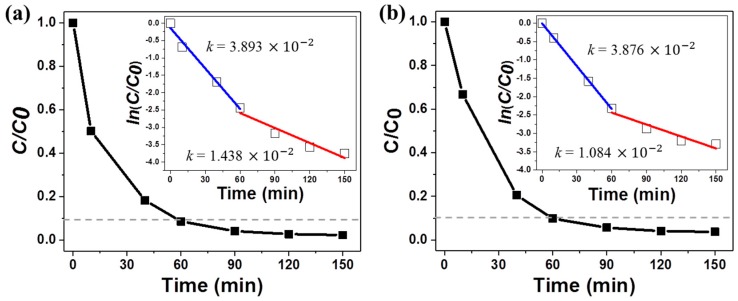
Photodegradation of an MB aqueous solution in the presence of 0.002 wt% TiO_2_/Ag NW composite nanomaterials under (**a**) UV-visible light and (**b**) visible light. The inset figures show the linear fits to *ln*(*C*/*C*_0_) vs. time.

**Figure 7 nanomaterials-09-00468-f007:**
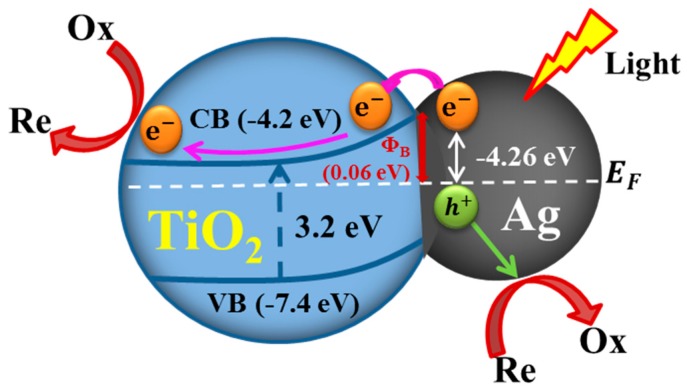
Schematic band energy diagram of charge transfer at the Ag NW/TiO_2_ NP interface (not drawn to scale). CB and VB indicate the conduction band and valence band of anatase TiO_2_. E_F_ presents the Fermi level of Ag. Φ_B_ is the Schottky barrier height.

**Figure 8 nanomaterials-09-00468-f008:**
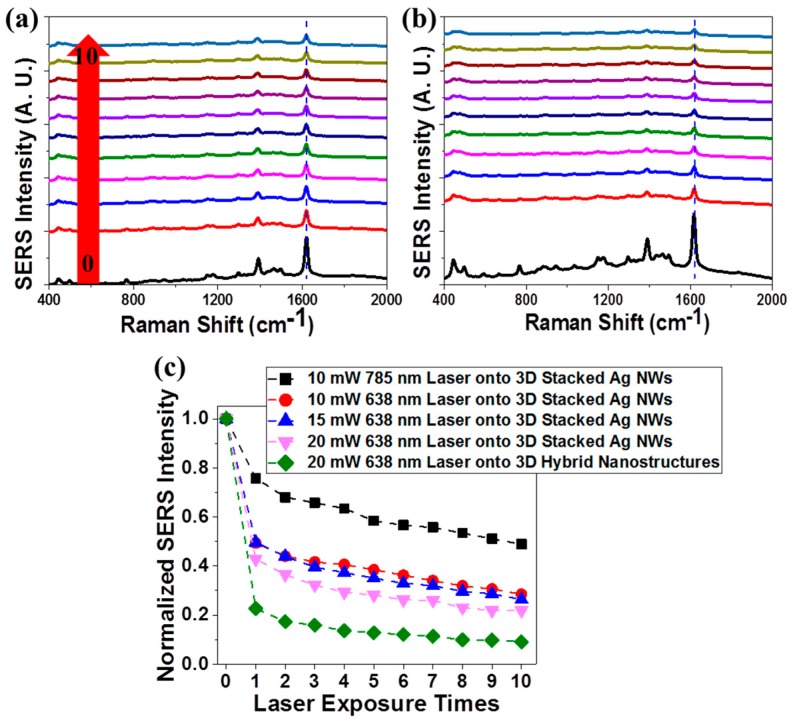
SERS intensity changes in the MB deposited onto (**a**) 3D stacked Ag NWs structures and (**b**) 3D hybrid (0.002 wt% TiO_2_/Ag NWs) nanostructures during 638 nm laser exposure (10 s). The SERS measurements were performed by illuminating with a 638 nm laser for only 0.01 s to avoid photocatalytic effects. (**c**) The overall SERS intensity changes at 1620 cm^−1^ in MB molecules under different illumination conditions.

**Table 1 nanomaterials-09-00468-t001:** Maximum and average field enhancement of crossed Ag NWs at different wavelengths.

Wavelength (nm)	Maximum Field Enhancement (|E|^2^/|E_0_|^2^)	Average Field Enhancement (|E|^2^/|E_0_|^2^)
450	2.9 × 10^4^	5.6 × 10^2^
532	3.2 × 10^3^	1.0 × 10^2^
633	4.3 × 10^2^	1.8 × 10^1^
785	8.6 × 10^1^	4.4 × 10^0^

**Table 2 nanomaterials-09-00468-t002:** The photocatalytic efficiencies of different photocatalysts under AM 1.5G and visible and NIR (the UV was cut off by a filter) light after 10 min illumination*.

Photocatalysts	UV-Visible	Visible
TiO_2_ 0.01 wt%/Microfibers*	14.3%	0.2%
3D Stacked Ag NWs/Microfibers*	36.3%	37.8%
0.01 wt% TiO_2_/3D Ag NWs Composite*	32.4%	23.7%
0.002 wt% TiO_2_/3D Ag NWs Composite*	49.8%	46%
0.001 wt% TiO_2_/3D Ag NWs Composite*	46.7%	45.8%

* 10 mL 0.05 mM MB was used to test the photodegradation of organic dyes during illumination.

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
