# Peer review of "Compact Integration of TiO2 Nanoparticles into the Cross-Points of 3D Vertically Stacked Ag Nanowires for Plasmon-Enhanced Photocatalysis"

_nanomaterials, 2019, doi:10.3390/nano9030468_

Round 1

Reviewer 1 Report

The paper reports on the realization of new composite nanomaterials consisting of TiO2 NPs compactly integrated into the 3D cross-points of vertically stacked Ag NWs. Their photocatalytic activity for photo degradation of organic dyes under UV-visible and visible illumination is tested.  

Supported by proper theoretical investigation, the paper demonstrates that photocatalytic activity is greatly improved due to the synergistic effects of visible light absorption by the 3D Ag NWs and efficient charge separation at the interface between the Ag NWs and TiO2 NPs.

The paper is sound,  motivations and discussion are well presented; I can say that it can be of great interest for the audience of the journal.

Just few comments:

- Information on the TiO2 NPs sizes are missing.

- For a better comprehension of the text, I would suggest to improve figures 2, 4 and 8. Figure 2 can be split in two figures or two different scales can be used for the two graphs otherwise details in extinction spectra are not clear. For example, as reported currently the sentence: “The 3D hybrid nanostructures possessed higher extinction than the 3D stacked Ag NWs” is evident  in the VIS region of the spectrum but it is not so clear at lower wavelengths. It is not clear the decrease in extinction intensity in that region and which peak is associated to LSPR resonance of metals.

As for figure 4, the authors say in the text: “the 3D stacked plasmonic Ag NWs exhibited higher photocatalytic activities compared to the TiO2 NPs “. This is not very clear in figure, as two similar colors are employed!

- The sentence “The average field enhancement corresponded to the maximum field enhancement (Table1) is not clear to me.

- I would suggest to report the wavelength regions in table 2 instead of “UV-Vis”, and “Vis”

-Finally, as for figure 8, apart suggesting an higher color contrast and improving the readability of the graph, it is not clear the temporal sequence of  the experiment: photodegradation process with 20mW laser was carried out before and than with a lower power laser on the same sample or figure 8 represents experiments on different samples? .

After clarified the above questions , the paper can be accepted for publication.

Author Response

Comment: The paper reports on the realization of new composite nanomaterials consisting of TiO2 NPs compactly integrated into the 3D cross-points of vertically stacked Ag NWs. Their photocatalytic activity for photo degradation of organic dyes under UV-visible and visible illumination is tested.  Supported by proper theoretical investigation, the paper demonstrates that photocatalytic activity is greatly improved due to the synergistic effects of visible light absorption by the 3D Ag NWs and efficient charge separation at the interface between the Ag NWs and TiO2 NPs. The paper is sound, motivations and discussions are well presented; I can say that it can be of great interest for the audience of the journal. Just few comments:

Comments 1: Information on the TiO2 NPs sizes are missing.

Response 1: The size of TiO2 NPs (~ 22 nm) was mentioned in the first subsection 3.1 (paragraph 1, line 10) of Results and Discussion part.

Comments 2: For a better comprehension of the text, I would suggest to improve figures 2, 4 and 8. Figure 2 can be split in two figures or two different scales can be used for the two graphs otherwise details in extinction spectra are not clear. For example, as reported currently the sentence: “The 3D hybrid nanostructures possessed higher extinction than the 3D stacked Ag NWs” is evident in the VIS region of the spectrum but it is not so clear at lower wavelengths. It is not clear the decrease in extinction intensity in that region and which peak is associated to LSPR resonance of metals. As for figure 4, the authors say in the text: “the 3D stacked plasmonic Ag NWs exhibited higher photocatalytic activities compared to the TiO2 NPs“. This is not very clear in figure, as two similar colors are employed!

Response 2: Regarding to Figure 2, the extinction spectra of 3D hybrid nanostructures and 3D stacked Ag NWs did not show much difference at lower wavelengths (Please see Figure S2 in Supporting Information) but clearly changed in visible region. We wanted to express the enhanced light-matter interactions of 3D hybrid nanostructures according to the standard solar illumination in the visible range. Therefore, we would keep the figure as it was and rewrite the sentence from “The 3D hybrid nanostructures possessed higher extinction than the 3D stacked Ag NWs” to “The 3D hybrid nanostructures possessed higher extinction than the 3D stacked Ag NWs in visible region” for better comprehension of the text in the revised manuscript.

As you recommended, Figures 4 and 8 were improved with higher color contrast and better readability of the graphs and edited in the revised manuscript.

Comments 3: The sentence “The average field enhancement corresponded to the maximum field enhancement (Table1) is not clear to me.

Response 3: As we stated in the original manuscript, the maximum field enhancement was obtained at the bottom corners of the central pentagonal Ag NW. The maximum field enhancement increased by an order of magnitude at shorter wavelengths (Table 1). The average field enhancement along the middle Ag NW surface could be extracted from the simulation results. We confirmed that both maximum and average filed enhancements increased by an order of magnitude at shorter wavelengths and were highly generated at the cross-points of the 3D stacked AgNWs.

Comments 4: I would suggest to report the wavelength regions in table 2 instead of “UV-Vis”, and “Vis”.

Response 4: Since the two different illumination ranges were determined by using a UV cut-off filter (not fixed at exact wavelength ranges), we used “UV-visible” and “Visible” to express the presence of UV light illumination. I appreciated your valuable comments.

Comments 5: Finally, as for Figure 8, apart suggesting a higher color contrast and improving the readability of the graph, it is not clear the temporal sequence of the experiment: photodegradation process with 20 mW laser was carried out before and than with a lower power laser on the same sample or Figure 8 represents experiments on different samples?

Response 5: Yes, you are right. The cycles consisted of a photodegradation process with high power laser and then a SERS measurement on the same spot of the substrate with a low power laser. For different photodegradation process, as presented in Figure 8c, the experiments were carried out on different samples. However, the SERS measurements conditions are same.

Reviewer 2 Report

The manuscript of Sung-Gyu Park and co-workers entitled “Compact Integration of TiO2 Nanoparticles into the Cross-Points of 3D Vertically Stacked Ag Nanowires for Plasmon-Enhanced Photocatalysis” describes the finding of new hybrid nanostructures based on semiconductor TiO2 nanoparticles (NPs) incoporated into the 3D crossed region of stacked plasmonic Ag nanowires (NWs). This is a very detailed and thorough study and materials are well characterized by different techniques, which suport the experimental evidences. The authors gathered sufficient proof for photocatalytic activity by SERS studies. This is also supported by theroretical simulations. The results of this work will be of great interest for the materials science community. Thus, I recommend the publication of this article  in Nanomaterials. My only comment to the authors is to check the spelling of some worrds as soome typos can be found along the manuscript.

Author Response

Comment: The manuscript of Sung-Gyu Park and co-workers entitled “Compact Integration of TiO2 Nanoparticles into the Cross-Points of 3D Vertically Stacked Ag Nanowires for Plasmon-Enhanced Photocatalysis” describes the finding of new hybrid nanostructures based on semiconductor TiO2 nanoparticles (NPs) incorporated into the 3D crossed region of stacked plasmonic Ag nanowires (NWs). This is a very detailed and thorough study and materials are well characterized by different techniques, which support the experimental evidences. The authors gathered sufficient proof for photocatalytic activity by SERS studies. This is also supported by theoretical simulations. The results of this work will be of great interest for the materials science community. Thus, I recommend the publication of this article in Nanomaterials. My only comment to the authors is to check the spelling of some words as some typos can be found along the manuscript. My only comment to the authors is to check the spelling of some words as some typos can be found along the manuscript.

Response: As you recommended, we checked some typos and revised the manuscript. We appreciated your valuable comments.

Reviewer 3 Report

The manuscript by  Linh et al. reports interesting results on plasmon-enhanced photocatalysis when combining Ag nanowires with TiO2. The combination of plasmonics and photocatalysis is a particularly promising research field, which can be enriched by potential implications of photothermal effects with novel applications that can arise.

The degree of innovation is good, as well as the potential impact.

I recommend publication of the manuscript, pending some revisions.

1) Previously, other researchers have studied the Ag/TiO2 hybrid system [1, 2]. These papers should at least mentioned.

2) The authors should consider that the presence of Ag nanoparticles implies also the presence of nanoscale thermal hotspot when irradiated with light, as shown in Refs. [3-12] These paper should be discussed.

3) These results are also particularly relevant for photocatalytic membranes. Authors should briefly mention this issue, with pertinent bibliography [8, 13-18].

4) Annotations in Figure 2 should be more readable

5) The “k” in the inset to Figure 6 should be far from the curve

Bibliography

[1] Photocatalytic Activity of Ag/TiO2 Nanotube Arrays Enhanced by Surface Plasmon Resonance and Application in Hydrogen Evolution by Water Splitting, Plasmonics 8 (2013) 501.

[2] Photocatalytic Reduction of NO with Ethanol on Ag/TiO2, Catal. Lett. (2010) 1.

[3] Overcoming temperature polarization in membrane distillation by thermoplasmonic effects activated by Ag nanofillers in polymeric membranes, Desalination 451 (2019) 192.

[4] Photothermal membrane distillation for seawater desalination, Adv. Mater. 29 (2017) 1603504.

[5] Thermoplasmonics: Quantifying Plasmonic Heating in Single Nanowires, Nano Lett. 14 (2014) 499.

[6] Thermoplasmonic effect of silver nanoparticles modulates peptide amphiphile fiber into nanowreath-like assembly, Nanoscale 7 (2015) 20238.

[7] Biomedical Applications of Shape-Controlled Plasmonic Nanostructures: A Case Study of Hollow Gold Nanospheres for Photothermal Ablation Therapy of Cancer, J. Phys. Chem. Lett. 1 (2010) 686.

[8] When plasmonics meets membrane technology, J. Phys.: Condens. Matter 28 (2016) 363003.

[9] Quantifying the efficiency of plasmonic materials for near-field enhancement and photothermal conversion, J. Phys. Chem. C 119 (2015) 25518.

[10] Thermo-plasmonics: using metallic nanostructures as nano-sources of heat, Laser Photonics Rev. 7 (2013) 171.

[11] Photoinduced heating of nanoparticle arrays, ACS Nano 7 (2013) 6478.

[12] Mapping Heat Origin in Plasmonic Structures, Phys. Rev. Lett. 104 (2010) 136805.

[13] Use of ultrafiltration membranes for the separation of TiO2 photocatalysts in drinking water treatment, Ind. Eng. Chem. Res. 40 (2001) 1712.

[14] Recent developments in photocatalytic water treatment technology: a review, Water Res. 44 (2010) 2997.

[15] Self-etching reconstruction of hierarchically mesoporous F-TiO2 hollow microspherical photocatalyst for concurrent membrane water purifications, J. Am. Chem. Soc. 130 (2008) 11256.

[16] Study on a photocatalytic membrane reactor for water purification, Catal. Today 55 (2000) 71.

[17] Studies on various reactor configurations for coupling photocatalysis and membrane processes in water purification, J. Membr. Sci. 206 (2002) 399.

[18] Heterogeneous photocatalytic degradation of pharmaceuticals in water by using polycrystalline TiO2 and a nanofiltration membrane reactor, Catal. Today 118 (2006) 205.

Author Response

Comment: The manuscript by Linh et al. reports interesting results on plasmon-enhanced photocatalysis when combining Ag nanowires with TiO2. The combination of plasmonics and photocatalysis is a particularly promising research field, which can be enriched by potential implications of photothermal effects with novel applications that can arise. The degree of innovation is good, as well as the potential impact. I recommend publication of the manuscript, pending some revisions.

Comments 1: Previously, other researchers have studied the Ag/TiO2 hybrid system [1, 2]. These papers should at least mentioned.

Response 1: As recommended, the papers were added and mentioned in the revised manuscript in the Introduction part (paragraph 2, line 5).

Comments 2: The authors should consider that the presence of Ag nanoparticles implies also the presence of nanoscale thermal hotspot when irradiated with light, as shown in Refs. [3-12] These papers should be discussed.

Response 2: As you commented, this issue was discussed again at the end of the first graph of subsection 3.5 in the revised manuscript. Please see the revised manuscript.

Comments 3: These results are also particularly relevant for photocatalytic membranes. Authors should briefly mention this issue, with pertinent bibliography [8, 13-18].

Response 3: As your recommended, some of the references were added and mentioned in the subsection 3.1 (paragraph 1, line 15) in the revised manuscript. Please see the revised manuscript.

Comments 4: Annotations in Figure 2 should be more readable.

Response 4: Figure 2 was edited in the revised manuscript.

Comments 5: The “k” in the inset to Figure 6 should be far from the curve.

Response 5: The positions of “k” were edited in the revised manuscript.

Reviewer 4 Report

The work of Linh et al., reports on the integration of TiO2 NPs for plasmonic application and photocatalysis. The authors explore the assembly with cross-point vertically stacked configuration to improve the efficiency of charge separation. They demonstrate high extinction across the UV range and support they results by FDTD simulations. The results are of interest as they show an emerging composite material for plasmon-enhanced photocatalysis with improved UV activity. The paper can be considered further for publication if the authors address a few minor points.

1-    The stability of Ag in solution is not addressed, do the authors noticed any effects related to the stability of Ag in solution. How would Fig. 8 change as function of time?

2-    Do the authors have any evidences, beside the simulations, in support of the photo-induced decomposition occurring at the cross-point areas? (physical analysis maybe ?)  

3-    Finally some previous work on Ag cross-point interaction, can be added [https://www.nature.com/articles/nmat3238]

Author Response

Comment: The work of Linh et al., reports on the integration of TiO2 NPs for plasmonic application and photocatalysis. The authors explore the assembly with cross-point vertically stacked configuration to improve the efficiency of charge separation. They demonstrate high extinction across the UV range and support they results by FDTD simulations. The results are of interest as they show an emerging composite material for plasmon-enhanced photocatalysis with improved UV activity. The paper can be considered further for publication if the authors address a few minor points.

Comments 1: The stability of Ag in solution is not addressed, do the authors noticed any effects related to the stability of Ag in solution. How would Fig. 8 change as function of time?

Response 1: Since Ag NWs are protected by surfactant molecules (i.e. poly(vinylpyrrolidone), PVP), we did not observe any stability issues of the Ag NWs in solution during 6 months. Figure 8 would exhibit in the same way as it was if the exposure time changed, i.e., the SERS intensities would significantly decrease after the first photodegradation process and remained unchanged during the next cycles, even when the exposure time gradually increase.

Comments 2: Do the authors have any evidences, beside the simulations, in support of the photo-induced decomposition occurring at the cross-point areas? (physical analysis maybe ?)

Response 2:  As the results shown in Figure 8, because Raman intensities are mainly contributed by hot spots (or cross-points of stacked Ag NWs) in SERS substrates, the significant decrease of SERS intensities indicated the decomposition of dye molecules presented at hot spot areas, which was pointed out as cross-point areas in the simulations results. Therefore, we would confirm that the photo-induced decomposition occurred at the cross-point regions.

Comments 3: Finally some previous work on Ag cross-point interaction, can be added. [https://www.nature.com/articles/nmat3238]

Response 3: The reference was added in the subsection 3.2 (paragraph 2, line 2) in the revised manuscript.
